# Long-Term Outcome of Dental Implants in Immediate Function Inserted on Autogenous Grafted Bone

**DOI:** 10.3390/jcm12010261

**Published:** 2022-12-29

**Authors:** Miguel de Araújo Nobre, Francisco Salvado, João André Correia, Maria Cristina Faria Teixeira, Francisco Azevedo Coutinho

**Affiliations:** 1Research, Development & Education, Maló Clinic, Avenida dos Combatentes 43, Level 11, 1600-042 Lisboa, Portugal; 2Clinica Universitária de Estomatologia, Centro Hospitalar Universitário Lisboa Norte—Hospital de Santa Maria, Avenida Professor Egas Moniz, 1649-028 Lisboa, Portugal; 3Faculdade de Medicina, Universidade de Lisboa, Avenida Professor Egas Moniz, 1649-028 Lisboa, Portugal; 4Departamiento de Ortodoncia, Facultad de Odontologia, Universidad Complutense de Madrid, Avenida de Séneca, 2, Ciudad Universitaria, 28040 Madrid, Spain

**Keywords:** alveolar ridge augmentation/methods, bone transplantation/methods, dental implants, immediate loading, immediate function, humans, retrospective studies, treatment outcome, long term

## Abstract

Background: There is a need for long-term evidence of immediate function dental implants inserted in grafted bone. The aim of this retrospective study was to investigate the outcome of full-arch rehabilitations supported by implants in grafted bone. Methods: Thirty-six patients (women: 24; men: 12; average age: 53.5 years) were included (225 implants). Primary outcome measure: to assess implant cumulative success rates evaluated through life tables. Secondary outcome measures: to evaluate implant and prosthetic survival, marginal bone loss, and the incidence of both biological and mechanical complications. Results: Twenty-five implants were unsuccessful giving a dental implant CS rate of 88.1% at 14 years and a 76.8% survival estimation (Kaplan–Meier) using the patient as the unit of analysis. No prosthesis was lost. Average MBL at 10 years was 2.01 mm. The incidence of biological complications was 36%, with smoking affecting it significantly (*p* < 0.001). The incidence of mechanical complications was 86.1% (45.2% and 54.8% in provisional and definitive prosthesis, respectively. Conclusions: The rehabilitation of atrophic maxillae through dental implants in immediate function inserted in grafted bone is a valid treatment alternative, despite the relevant rate of implant failures and incidence of complications.

## 1. Introduction

Oral disorders, including edentulism, are considered a heavy burden for the world population, affecting 44.5% of the global population [1]. Edentulism is responsible for the largest proportion share of the global estimated burden, with an estimated 100,000 of the 275,000 disability-adjusted life years due to oral cancer, oral diseases, and other disorders [2]. Furthermore, the alveolar bone resorption occurring after tooth loss implies a serious limitation to rehabilitate with an implant-supported fixed prosthesis [3,4].

Bone grafting is one of the alternatives to overcome the limitation of lack of bone volume for implant-supported restorations [5]. The most common reconstructive techniques include maxillary sinus floor augmentation and onlay grafts, both providing stable baseline conditions for implant insertion with survival rates of 86% (onlay grafts up to 5 years) [6] and 88.6% to 100% (maxillary sinus floor augmentation) [7]. Considering this alternative, autogenous bone grafting is recognized as the gold standard given its characteristics including osteogenesis, osteoinductiveness, and osteoconductiveness [8], while allowing large amounts of both cortical and cancellous bone to be harvested [9,10]. Nevertheless, some disadvantages include post-operative patient discomfort, sensitivity, and pain in the donor site [11].

The restoration of edentulism in the atrophic maxilla through implant-supported fixed prosthesis in grafted bone requires a multi-step process: First, maxilla reconstruction with an autogenous bone graft harvested from the iliac crest; second, immediate provisional prosthetic rehabilitation; and third, rehabilitation with a fixed bridge supported by immediate-function implants, 6 months after the graft procedure [11]. To allow the possibility of a fixed oral rehabilitation supported by immediate function implants (third step), it is necessary to achieve graft stability and volume while satisfying minimal prosthetics rehabilitation conditions during the healing phase. A previous study [11] investigated different methods to provide immediate prosthetic rehabilitation during the first step while preserving graft stability (avoiding compression by the prosthesis) in edentulous maxilla: a fixed prosthesis supported by titanium dental implants placed in non-grafted bone, a fixed prosthesis supported by residual natural teeth, a removable prosthesis supported by titanium palatal dental implants (acting as a pseudo-scaffold), or a removable prosthesis with palatal mucosa retention. All methods provided bone graft stability to allow the restoration with implant-supported fixed prosthesis after 6 months of the grafting procedure, enabling an implant cumulative survival rate of 96.7% after 5 years of follow-up. The long-term outcome of implants placed in grafted bone is a topic that receives great attention from the scientific community, with survival rates ranging between 75% and 95% [12,13,14,15].

The aim of this study was to report the long-term outcomes (14 years clinical and 10 years radiographic) of dental implants inserted in grafted bone with immediate function for the full-arch rehabilitation of the maxillae.

## 2. Materials and Methods

This prospective clinical study was performed complying with all ethical regulations in accordance with the Declaration of Helsinki and approved by an Ethical Committee (Ethical Committee for Health, Lisbon, Portugal; authorization no. 009/2017. The study was carried out in a private practice in Lisbon, Portugal, with an adequate understanding and after obtaining written consent from all the included subjects.

### 2.1. Inclusion and Exclusion Criteria

The inclusion criteria were patients requiring a bone graft procedure for total reconstruction of the maxilla (bilateral full sinus-lift and maxilla onlay grafts) with the objective of placing endosseous implants in immediate function to support a full-arch fixed prosthetic implant-supported rehabilitation. As exclusion criteria, patients not followed by our team, patients considered emotionally unstable or submitted to active maxillary radiation therapy were excluded from the study.

Thirty-six patients were included consecutively (men: 13, women: 24), with an average age of 53.5 years (range: 32–72 years). The maxilla reconstruction procedures were performed between August 1997 and October 2005, with the patients receiving a maxillary reconstruction with onlay grafts (for width augmentation) and sinus lift (for bone height augmentation). Usually, six months after the graft procedure, the implants were inserted in immediate function (between November 1999 and May 2006). The patients were followed for 14 years after the edentulous maxilla prosthetic rehabilitation with immediate-function implants.

### 2.2. Surgical and Prosthetic Protocols

The rehabilitation followed a 3-step process with maxilla reconstruction with autogenous bone graft harvested from the iliac crest; followed by immediate provisional prosthetic rehabilitation for the following 6-months of follow-up; and finalized with the insertion of dental implants in immediate function on grafted bone [11].

For the bone grafts, harvesting from the internal aspect of the anterior border of the iliac crest was performed, removing a block with cortical and spongeous bone. Considering the bone height augmentation, a modified sinus-lift technique was performed, with a window opening following the maxilla’s residual crest and limited posteriorly by the first molar and anteriorly by the anterior wall of the sinus, with 20 mm in height from the residual crest. Bone blocks usually 10 × 10 mm were inserted in the sinus and stabilized either by pressure or osteosynthesis screws, with spaces filled in with spongeous bone. To increase the residual’s crest width, onlay grafts were performed.

The immediate temporary prosthetic rehabilitation was divided inti four groups according to the patients’ characteristics and considering the conditions for stabilization of the grafts during the healing phase: Group 1—implant-supported fixed prosthesis (n = 10 patients; n = 38 immediate function implants placed in nongrafted bone, not included in the study) consisting of a full-arch high-density acrylic-resin prosthesis with titanium cylinders; Group 2—mucosa-retained removable prosthesis (n = 7 patients with palate anatomy presenting good retention) with relining of the patient’s acrylic dentures; Group 3—tooth-retained fixed prosthesis (n = 6 patients) with an acrylic resin prosthesis supported by the patients’ natural teeth; Group 4—palatal implant retained (n = 13 patients; n = 17 palatal implants, MkIV 4 × 7 mm, n = 17 multi-unit abutments, Nobel Biocare AB) with acrylic resin prosthesis trimmed and relined to fit the multi-unit abutments (Table 1).

The insertion of dental implants (n = 225 implants: n = 17 MkIII implants; n = 95 MkIV implants; n = 113 NobelSpeedy implants; Nobel Biocare AB) in immediate loading on grafted bone was performed on average 6 months after the bone grafting procedure. A mucoperiosteal flap was raised along the top of the ridge with relieving incisions in the buccal aspect of the molar area. The osteosynthesis screws were removed and the stability of the bone grafts was assessed. The insertion of the implants followed standard procedures [16], except the employment of underpreparation to achieve a final insertion torque ≥ 30 Ncm. The implant necks were aimed to be positioned at the bone level and connected to abutments (Estheticone; Miruscone; Multi-unit; Nobel Biocare AB). The flap was closed with a 3-0 nonresorbable suture. A new provisional full-arch high-density, heat–processed, all-acrylic prostheses (acrylic resin gingiva and prosthetic teeth; Heraeus Kulzer GmbH) was connected to the implants on the day of surgery to replace the temporary prosthetic rehabilitation that functioned during the grafts healing phase.

The definitive prosthesis was connected on average, 6 months after the implant surgery. For the definitive prosthesis, a metal-ceramic implant-supported fixed prosthesis with a titanium framework and all-ceramic crowns (Maló Clinic Ceramic Bridge; NobelProcera titanium framework, NobelProcera crowns, Nobel Rondo ceramics; Nobel Biocare AB), or a metal-acrylic resin implant-supported fixed prosthesis with a titanium framework (Maló Clinic Acrylic Bridge; NobelProcera titanium framework; Nobel Biocare AB) and acrylic resin prosthetic teeth (Heraeus Kulzer GmbH), were used to replace the provisional prosthesis, taking into consideration the patient’s desire.

The patients were enrolled in a maintenance protocol with clinical evaluations and prophylaxis performed at post-operative 10 days, 2-, 4-, and 6-months, 1 year, and thereafter every 6 months until 14 years of follow-up. A clinical case is illustrated in Figure 1, Figure 2 and Figure 3.

### 2.3. Outcome Measures

An outcome assessor blinded to the objectives of the study evaluated the data. Outcomes were assessed over a 14-years follow-up. Primary outcome measure was implant success, based on the success criteria adopted by the authors [17]: (a) implant fulfilled its intended function supporting the reconstruction (sleeping implants were considered failures); (b) implant was stable upon manual testing; (c) absence of persistent infection jeopardizing the implant outcome; (d) absence of radiolucency around the implants; (e) good aesthetic result (classified as the absence of aesthetic complains from the patient and the prosthodontist); and (f) allowed construction of an implant-supported fixed restoration that was comfortable for the patient and permitting good hygienic maintenance (classified as the absence of complaints from the patient and the prosthodontist). Implants not complying with the criteria were considered survivals. Implant removal was classified as a failure.

Secondary outcome measures were prosthetic survival (based on function with the necessity of replacing the prosthesis classified as failure), marginal bone loss at 10 years, the incidence of mechanical complications, and the incidence of biological complications.

The radiographic evaluation to assess marginal bone loss was performed at baseline (implant surgery) and 10 years of follow-up using periapical radiographs through the parallelometric intraoral technique. For the intraoral technique, a conventional radiograph holder was used, the position of which was adjusted manually to ensure orthogonal film positioning. A blinded operator examined all radiographs of the implants for marginal bone level. Each periapical radiograph was scanned at 300 dpi with a scanner (HP Scanjet 4890, HP Portugal, Paço de Arcos, Portugal). The marginal bone level was assessed with image analysis software (Image J version 1.40 g for Windows, National Institutes of Health, Bethesda, MD, USA) using the implants’ inter-thread distance as a reference for digital calibration. The implant platform was used as a reference point and marginal bone loss was defined as the difference in marginal bone levels between the day of surgery and the point of evaluation. The radiographs were accepted or rejected for evaluation based on the clarity of the implant threads; a clear thread guarantees both sharpness and an orthogonal direction of the radiographic beam towards the implant axis.

The following mechanical complication factors were assessed: fracture or loosening of mechanical and prosthetic components. The following biological complication factors were assessed: peri-implant pathology (defined as peri-implant pocket depths ≥5 mm, bleeding on probing, with concurrent marginal bone loss compared to the previous radiograph or clinical attachment loss of >2 mm) [18], fistula formation, or abscess.

### 2.4. Statistical Analysis

Descriptive statistics (average, standard deviation, range) were calculated for age and marginal bone loss (at 10 years). Frequencies were used to classify biological complications, loss to follow-up, and prosthetic survival. Inferential analysis was performed to evaluate the difference in demographics between patients with complete follow-up and patients lost to follow-up (age: Mann–Whitney U test; sex: chi-square test). Cumulative implant survival and success were estimated at the patient level (any implant failure in each patient) through the Kaplan–Meier product limit estimator (with a log-rank test to compare survival curves) and at the implant level using life tables. The comparison of marginal bone loss and biological complications distribution was performed using the chi-square test, Student’s T-test, and Mann–Whitney U test. The significance level was set at 5%. Data were statistically analyzed using the Statistical Package for the Social Sciences software (IBM SPSS, version 17, Rochester, NY, USA).

## 3. Results

Four patients were lost to follow-up (11.1%). Of the 36 patients enrolled, there were 12 patients who were smokers and 12 patients with systemic conditions (Table 1); with 4 patients with more than one condition.

A total of 225 implants were inserted in 36 patients (Table 1 and Table 2). Implant failures occurred in 8 patients, giving an overall cumulative survival estimation of 76.8% (Table 2 and Table 3, Figure 4; Kaplan–Meier) after 14 years of follow-up using the patient as the unit of analysis (first implant failure in any patient censored, independently of the remaining implants maintaining function; this evaluation displays the cumulative percentage of patients that did not experience implant failure). No significant differences in implant survival were found for smokers, systemically compromised patients, and between prosthetic groups, nor for age and sex.

A total of 15 implant failures occurred (Table 2), giving an overall implant cumulative survival rate of 93.0% at 14 years of follow-up (Table 4, Figure 5).

Despite the implant failures, all prostheses remained in function. The overall average (standard deviation) marginal bone loss at 10 years was 2.01 mm (1.58 mm) (Table 5).

Mechanical complications occurred in 31 patients (86.1%), with 45.2% occurring in the provisional prostheses (n = 14 patients) and the remaining in the definitive prostheses (n = 17 patients; 54.8%). The complications of the provisional prosthesis were: Abutment screw loosening in 7 patients (22.6%), and fracture of the provisional prosthesis in 10 patients (32.3%). The complications of the definitive prosthesis were: abutment screw loosening (n = 3 patients); abutment fracture (n = 1 patient); fracture of acrylic resin crowns (n = 6 patients and 7 crowns); fracture of ceramic crowns (n = 6 patients and 14 crowns). These complications were resolved by re-tightening the abutments/prosthetic screws, repairing the acrylic-resin or metal-acrylic resin prostheses, replacing the ceramic crowns in metal-ceramic prostheses, adjusting the occlusion, and manufacturing a night guard. No further mechanical complications were observed. Biological complications occurred in 13 patients (36%), consisting of peri-implant pathology (n = 12 patients, 33.3%; n = 21 implants) and suppuration (n = 1 patient, 2.8%, n = 1 implant). Peri-implant pathology was resolved in 6 patients and 9 implants (5 patients and 8 implants with peri-implant pathology; 1 patient with suppuration in 1 implant) by non-surgical intervention (cleaning the peri-implant pocket using the ultrasonic scaler and irrigating the pocket with 0.2% chlorhexidine gel); while one patient had one implant resolved through surgical intervention and one implant which the intervention did not resolve the complication; and in six patients and 11 implants, peri-implant pathology was not resolved (in one patient, 2 implants were removed) despite the surgical intervention attempted to resolve the condition. A significant difference between smokers and non-smokers was registered in the incidence of biological complications (*p* < 0.001, chi-square test), while no significant differences were registered for sex nor systemic condition.

Considering the success criteria and the number of implants with persistent biological complications, the implant cumulative success rate was 88.1% at 14 years of follow-up (Table 6, Figure 6).

## 4. Discussion

The overall 93% cumulative survival and 88.1% cumulative success outcomes of implants inserted in immediate function on grafted bone at 14 years registered in the present study provide evidence of predictability for this treatment modality. To the authors’ knowledge, this study constitutes the longest follow-up recorded for dental implants with immediate function inserted in grafted bone. It compares favorably to other reported immediate/early loading protocols for edentulous maxilla rehabilitation: Mordenfeld et al. [12] reported an 86% implant survival rate at 10 years of follow-up; Soehardi et al. [13] registered a 74.8% implant survival rate at 10 years of follow-up; de Morais et al. [14] reported an 88.7% implant survival rate from 8–10 years of follow-up; and Nyström et al. [19] reported 90% cumulative implants survival rates from 9–14 years. These results suggest that the technique for reconstruction of the maxilla is adequate, warranting good conditions for implant rehabilitation that can be maintained over long-term follow-ups. Nevertheless, it is not an uneventful rehabilitation process, with a significant number of patients (over 20%) experiencing implant failure. Despite the non-significant difference, it is noteworthy to point out that most implant failures occurred in women (7/8 patients, 87.5%) and in smokers (4/8 patients, 50%), following a similar pattern previously reported [13]. The implant survival and success outcomes of the present study were generally lower when compared with other long-term studies reporting on immediate function dental implants placed in non-grafted bone with one exception, where 81.9% survival and 75.7% success rates were registered at 11 years [20]. Generally, for maxillary dental implants placed in immediate function for full-arch rehabilitation, survival ranged between 94.7% and 97.5% [21,22], whereas a 93.9% success rate was registered [22] between 10 and 13 years of follow-up. A recent systematic review and meta-analysis to investigate the outcome of fixed full-arch rehabilitations supported by tilted and axial implants further highlighted this difference, registering a 96.6% survival for maxillary dental implants at 10 years of follow-up [23].

The average marginal bone loss of 2 mm at 10 years is considered a stable outcome, representing a stable condition maintained throughout the follow-up. Nevertheless, a heterogenous result was observed, with 23% and 16% of the implants exhibiting over 2 mm and 3 mm of marginal bone loss, respectively. A significant proportion of the implants with over 3 mm of marginal bone loss (15/28, 53.6%) can be explained by the incidence of peri-implant pathology and/or smoking habits [13], while for the remaining 46.4% (n = 13 implants) no potential risk indicators were observed. This result finds a parallel in the literature where a similar marginal bone loss was registered for implants inserted in grafted bone for full-arch maxillary reconstructions, ranging between 1.6 mm and 3.1 mm at 10 years [12,14,19,24]. Nevertheless, the present study is the only one representing implants inserted in immediate function. The marginal bone loss reported in long-term studies evaluating the outcome of dental implants inserted in non-grafted bone ranged between 0.79 mm and 1.67 mm between 10 and 11 years of follow-up [20,21,22]. When compared to these studies, an increased bone loss was noted for the present study as expected.

The 36% incidence of biological complications impacted the long-term outcome negatively in both implant survival and marginal bone loss. Smoking exerted a significant effect, with 9 of the 13 smokers registering biological complications. Smoking has long been described in the literature as a risk indicator for both biological complications and marginal bone loss, exerting significant influence on the outcome of dental implants as reported in multiple systematic reviews and meta-analyses [25,26,27].

The incidence of mechanical complications was very high, occurring in 86.1% of the patients (31/36 patients), a situation that may be related to the fact that in 28 patients an implant-supported fixed prosthesis was present as opposing dentition. Previous reports registered the presence of opposing implant-supported prostheses as a risk indicator for mechanical complications, implying a potential lack of shock absorption from the prosthesis or proprioception from the patient [28,29]. A further potential explanation could be the presence of bruxing habits [28,29]; however, the authors were not able to collect information for this variable prior to the maxillary reconstruction and therefore stands as a limitation for the interpretation of this result.

The dropout rate was low (n = 4 patients, 11.1%) and accounts for a strength in the study’s internal validity. Adding to the previously mentioned limitation in the absence of information on patients’ bruxing habits, further limitations of this study include a single center, the small sample size, and the predominance of women over men in the sample, which suggests caution in the generalization of the results.

Future research should focus on the comparison between the current protocol and other alternative procedures (such as zygomatic implants) on the long-term outcome and incorporating patient satisfaction evaluation measures.

## 5. Conclusions

Within the limitations of this study, it can be concluded that the rehabilitation of the severely atrophied maxilla through dental implants inserted in grafted bone with immediate function is a valid treatment alternative in the long term. Prosthetic survival was high and cumulative implant survival was acceptable, with stable marginal bone loss. Nevertheless, a relevant rate of patients experienced implant failure, with increased incidence of biological and mechanical complications. Smoking exerted a significant effect on the incidence of biological complications.

## Figures and Tables

**Figure 1 jcm-12-00261-f001:**
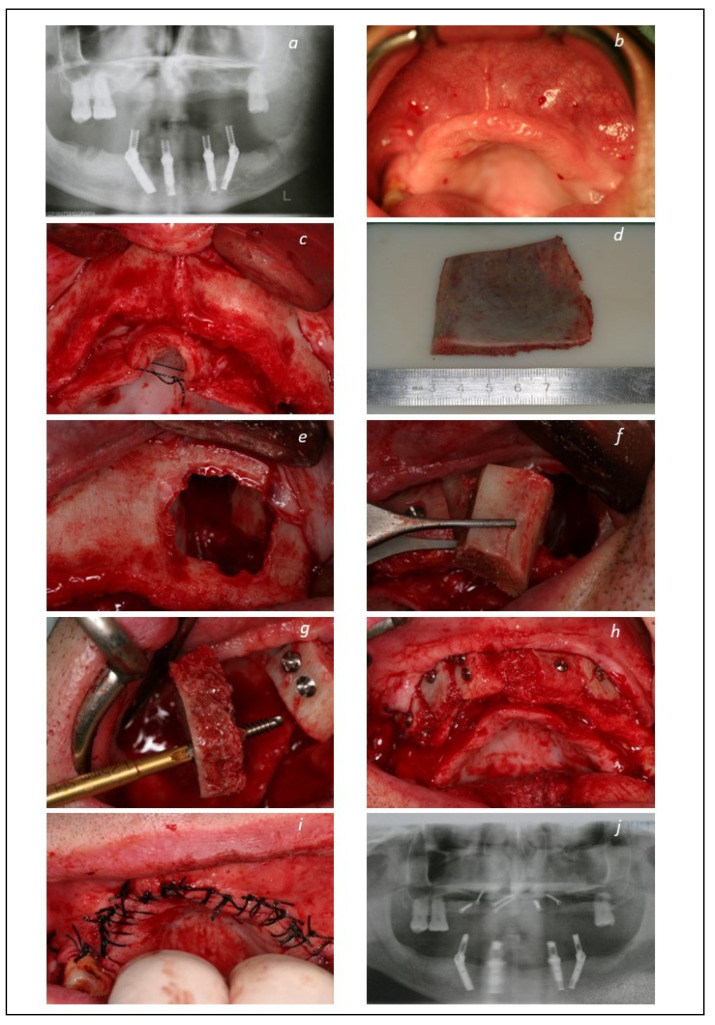
Maxillary reconstructive procedure: (**a**) Pre-operative orthopantomography; (**b**) intra-oral pre-operative photograph of the maxillary ridge; (**c**) intra-oral per-operative photograph exhibiting the maxillary atrophic ridge; (**d**) cortical/spongeous bone fragment harvested from the iliac crest; (**e**) intra-oral photograph of the maxillary sinus; (**f**) intraoral photograph of the cortical bone fragment placement in the sinus; (**g**) intraoral photograph of the onlay graft cortical bone fragment placement; (**h**) intraoral photograph of the onlay graft completed including cortical bone fragments attached to the maxillary residual crest and spongeous bone filling the gaps; (**i**) intraoral photograph of the closed flap; (**j**) post-operative orthopantomography after bone graft procedure.

**Figure 2 jcm-12-00261-f002:**
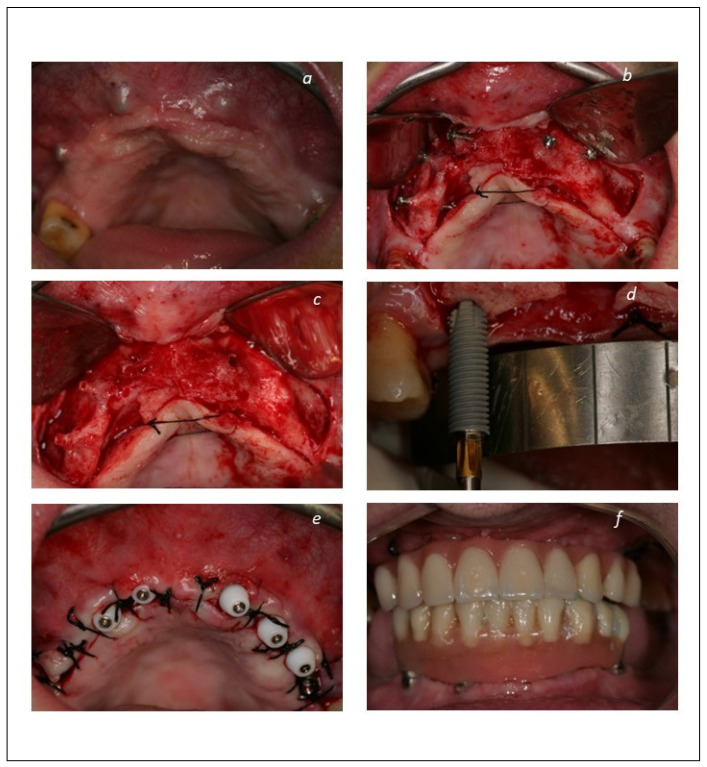
Maxillary rehabilitation procedure with implants in immediate function: (**a**) Intraoral photograph of the maxillary arch 6 months after the bone graft procedure and on the day of implant rehabilitation; (**b**) intraoral photograph of the maxillary ridge on the day of implant rehabilitation. Note the increased volume of the maxilla enabling implant insertion; (**c**) intraoral peroperative photograph after removal of the osteosynthesis screws and prior to implant insertion; (**d**) intraoral peroperative photograph illustrating implant insertion in grafted bone; (**e**) intraoral post-operative oclusal view with the implants/abutments protected with healing caps to prevent the soft tissue from collapsing while manufacturing the prosthesis at the dental laboratory; (**f**) intraoral post-opertative photograph with the acrylic resin prosthesis connected on the day of surgery and achieving immediate function.

**Figure 3 jcm-12-00261-f003:**
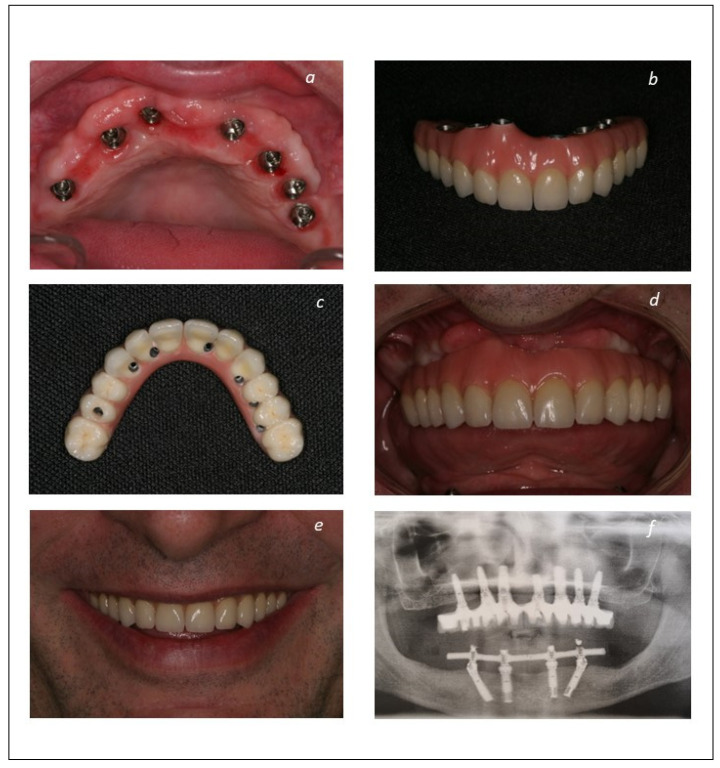
Definitive prosthodontic rehabilitation and follow-up: (**a**) Intraoral photograph 6 months post-loading of the implants at the connection of the final prosthesis; (**b**) frontal view of the final metal-ceramic prosthesis (Maló Clinic Ceramic Bridge—titanium infrastructure and ceramic crowns cemented individually); (**c**) occlusal view of the final metal-ceramic prosthesis (Maló Clinic Ceramic Bridge); (**d**) intraoral frontal view of the final metal-ceramic prosthesis (Maló Clinic Ceramic Bridge); (**e**) patient smiling with the final metal-ceramic prosthesis (Maló Clinic Ceramic Bridge); (**f**) orthopantomography of the rehabilitation at 10 years of follow-up.

**Figure 4 jcm-12-00261-f004:**
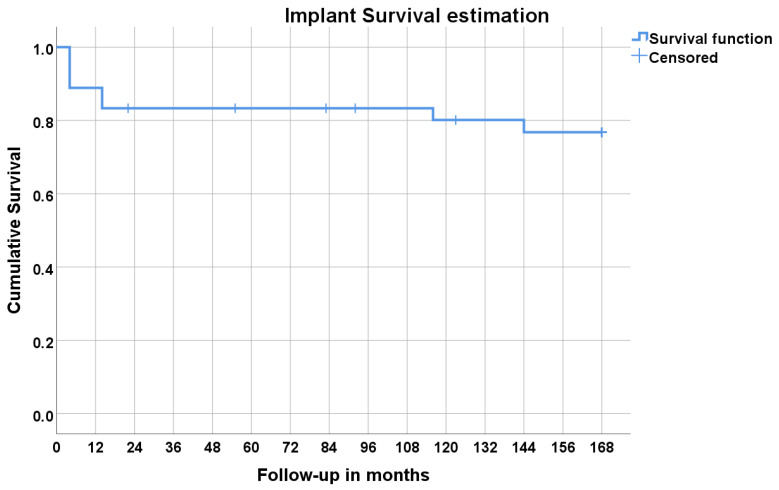
Survival estimation using the Kaplan–Meier product limit estimator considering the patient as unit of analysis (the first implant failure was censored independently of the remaining implants maintaining function; this evaluation displays the cumulative percentage of patients that did not experience implant failure).

**Figure 5 jcm-12-00261-f005:**
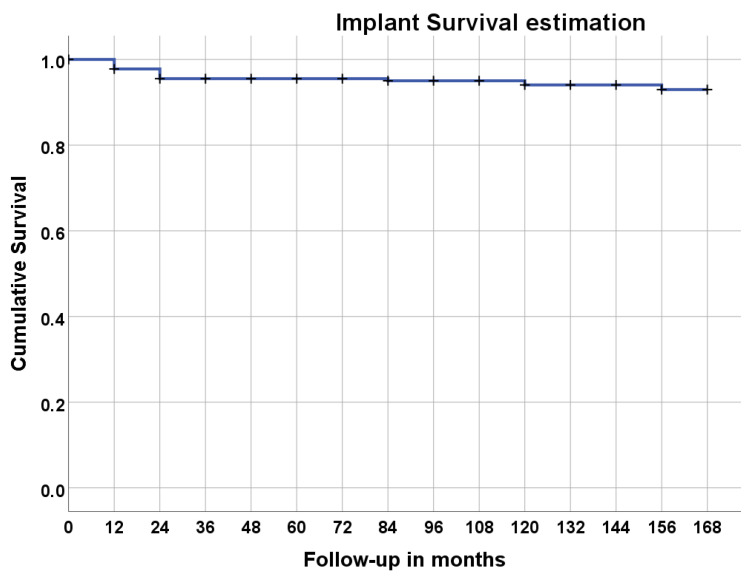
Implant cumulative survival rate at 14 years of follow-up.

**Figure 6 jcm-12-00261-f006:**
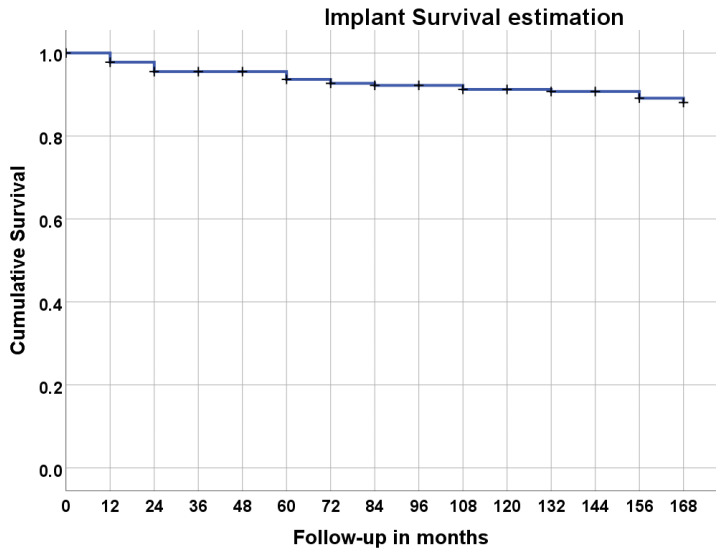
Cumulative implant success rate.

**Table 1 jcm-12-00261-t001:** Sample characteristics according to the immediate temporary prosthetic group distribution.

	Total Sample	Group 1 (Implant-Supported Fixed Prosthesis)	Group 2 (Mucosa-Retained Removable Prosthesis)	Group 3 (Tooth-Retained Fixed Prosthesis)	Grupo 4 (Palatal-Implant Retained Prosthesis)
Number of patients (male/female)	36 (12/24)	10 (6/4)	7 (0/7)	6 (4/2)	13 (2/11)
Average age in years	54	48	55	54	57
Smoking habits (n patients)	12	5	1	1	5
Systemic condition (n patients)	12 *	2	3	2	5
Cardiovascular condition	9	2	1	2	4
Thyroid condition	2	1	1	0	0
Rheumatologic condition	2	0	2	0	1
Oncologic condition	1	0	0	0	1
Inflammatory condition	1	0	0	0	1
Number of implants inserted at step 1 (bone graft) in non-grafted bone	55	38	0	0	17 (palatal)
Number of failed grafts	0	0	0	0	0
Number of implants inserted in immediate loading on grafted bone	225	48	48	39	90

* 4 patients with more than one condition; total 12 patients.

**Table 2 jcm-12-00261-t002:** Implant distribution in the sample and failure analysis.

**Implant Distribution according to Type**				
**Type**	**Number of Implants (Number of Failures)**			
Mk III	17 (0)			
Mk IV	95 (11)			
NobelSpeedy Groovy	113 (4)			
Total	225 (15)			
**Implant failure analysis**				
**Patient no. and characteristics**	**Implant position**	**Implant type**	**Follow-up in months**	**Reason for failure**
1 (male, 45y) Cardiovascular disease; smoker	#21	MkIV	8	Loss of osseointegration
2 (female, 49y)	#11	MkIV	22	Loss of osseointegration
3 (female, 70y)	#21	MkIV	4	Loss of osseointegration
	#23	MkIV	12	
	#11	MkIV	23	
4 (female, 62y) Heavy bruxer	#12	MkIV	4	Loss of osseointegration
	#23	MkIV	73	
5 (female, 56y) Cardiovascular disease; smoker	#23	MkIV	144	Loss of integration
	#25	MkIV	144	Peri-implant pathology
6 (female, 49y) Cardiovascular disease; smoker	#24	MkIV	4	Loss of osseointegration
	#21	MkIV	5	
7 (female, 65y) Smoker	#25	NobelSpeedy Groovy	116	Peri-implant pathology
	#15	NobelSpeedy Groovy	117	
8 (female, 35y)	#14	NobelSpeedy Groovy	14	Loss of osseointegration
	#21	NobelSpeedy Groovy	14	

**Table 3 jcm-12-00261-t003:** Cumulative implant survival estimation at 14 years using patient as the unit of analysis (Kaplan–Meier product limit estimator).

Time (Months)	Status (0 = non-Failure, 1 = Failure *)	Cumulative Proportion Surviving		Cumulative Events (n)	Patients at Risk (n)
		Estimate	Standard Error		
0	0			0	36
4	1	0.889	0.052	4	32
12	0			4	32
14	1	0.833	0.062	6	30
24	0			6	29
36	0			6	29
48	0			6	29
60	0			6	28
72	0			6	28
84	0			6	27
96	0			6	26
108	0			6	26
116	1	0.801	0.067	7	25
120	0			7	25
132	0			7	24
144	1	0.768	0.072	8	23
168	0			8	23

* Failure was defined as the first implant failure in a patient irrespective of the remaining implants maintaining function. This evaluation displays the cumulative percentage of patients that did not experience implant failure.

**Table 4 jcm-12-00261-t004:** Cumulative survival rate of trans-sinus implants at implant level.

Duration	Total Implants	Failed	Lost to Follow-Up	Survival Rate %	Cumulative Survival Rate%
Placement–1 year	225	5	0	97.8%	97.8%
1 year–2 years	220	5	8	97.7%	95.5%
2 years–3 years	207	0	0	100%	95.5%
3 years–4 years	207	0	0	100%	95.5%
4 years–5 years	207	0	6	100%	95.5%
5 years–6 years	201	0	0	100%	95.5%
6 years–7 years	201	1	6	99.5%	95.0%
7 years–8 years	194	0	0	100%	95.0%
8 years–9 years	194	0	0	100%	95.0%
9 years–10 years	194	2	0	99.0%	94.1%
10 years–11 years	192	0	12	100%	94.1%
11 years–12 years	180	0	6	100%	94.1%
12 years–13 years	174	2	0	98.9%	93.0%
13 years–14 years	172	0	0	100%	93.0%

**Table 5 jcm-12-00261-t005:** Marginal bone loss at 10 years of follow-up.

Mean (mm)	2.01	
SD (mm)	1.58	
Number	176	
Frequencies	N	%
0 mm	9	5.1%
0.1–1.0 mm	37	21.0%
1.1–2.0 mm	62	35.2%
2.1–3.0 mm	40	22.7%
>3.0 mm	28	15.9%

**Table 6 jcm-12-00261-t006:** Cumulative success rate of the implants at 14 years.

Duration	Total Implants	Unsuccessful	Lost to Follow-Up	Survival Rate %	Cumulative Survival Rate%
Placement–1 year	225	5	0	97.8%	97.8%
1 year–2 years	220	5	8	97.7%	95.5%
2 years–3 years	207	0	0	100%	95.5%
3 years–4 years	207	0	0	100%	95.5%
4 years–5 years	207	4	6	98.0%	93.6%
5 years–6 years	197	2	0	99.0%	92.7%
6 years–7 years	195	1	6	99.5%	92.2%
7 years–8 years	188	0	0	100%	92.2%
8 years–9 years	188	2	0	98.9%	91.2%
9 years–10 years	186	0	0	100%	91.2%
10 years–11 years	176	1	9	99.4%	90.7%
11 years–12 years	170	0	6	100%	90.7%
12 years–13 years	167	3	0	98.2%	89.1%
13 years–14 years	165	2	0	98.8%	88.1%

## Data Availability

Data are available from a public repository, https://osf.io/qkgac, last accessed on 25 November 2022.

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
