# Peer review of "Long-Term Outcome of Dental Implants in Immediate Function Inserted on Autogenous Grafted Bone"

_jcm, 2022, doi:10.3390/jcm12010261_

Round 1

Reviewer 1 Report

I want to thank the authors for their article; i feel some slight changes should be made in order to improve the overall quality of the final publication.

1.     I would specify that the “immediate restoration” provided in step 2 is a provisional one as, given the more broadly use of the term “immediate” in “immediate loading” that could be confusing to readers.

2.     Why were block grafts also used for the sinus lift, considering the good results reported in the literature with other materials?(such as DBBM)

3.     Line 121: change with “immediate loading”

4.     The authors should better explain the prosthetic phases that lead to the fixed provisional and definitive prosthesis.

5.     What implants brands was used? (maybe Nobel Biocare but it is not made completely clear)

6.     I would argue that criteria e) and f) are not particularly objective and may hamper the final results.

7.     “Implant lost integration after abutment fracture”: what’s the meaning behind this phrase?

8.     Line 257: I would eliminate this phrase altogether an implant failure is obligatory cause of the failure of the related prostheses.

9.     Line 275: Authors should specify if those complications were mucositis or peri-implantitis.

Author Response

The authors thank the Reviewer for all the effort put into the analysis of our manuscript. Please find below the point-by-point response to the review.

I want to thank the authors for their article; i feel some slight changes should be made in order to improve the overall quality of the final publication.

  1. I would specify that the “immediate restoration” provided in step 2 is a provisional one as, given the more broadly use of the term “immediate” in “immediate loading” that could be confusing to readers.

Response: The authors thank the Reviewer’s suggestion. The specification was introduced as requested.

Changes: Lines 53,54,99,110

  1. Why were block grafts also used for the sinus lift, considering the good results reported in the literature with other materials?(such as DBBM).

Response: The authors thank the Reviewer’s query. The maxillary reconstructions were performed some time ago, and the surgeon had experience with the method reported and therefore we did not wish to add another layer of complexity with the learning curve needed to master a different technique. It was a purely clinical driven decision process.

Changes: None.

  1. Line 121: change with “immediate loading”

Response: The authors thank the Reviewer’s indication. The term was changed as requested.

Changes: Lines 124,125

  1. The authors should better explain the prosthetic phases that lead to the fixed provisional and definitive prosthesis.

Response: The authors thank the Reviewer’s indication. A more thorough explanation was provided as requested.

Changes: Lines 110,132,134,135

  1. What implants brands was used? (maybe Nobel Biocare but it is not made completely clear)

Response: The authors thank the Reviewer’s query. The Reviewer is correct, it was Nobel Biocare implants. The authors made it clear in the manuscript.

Changes: Lines 123,124

  1. I would argue that criteria e) and f) are not particularly objective and may hamper the final results.

Response: The authors thank the Reviewer’s indication. The criteria are usually used by the authors. Despite the potential for subjectiveness, these criteria are collected using the patient and prosthodontist evaluations. The authors included this information in the manuscript ot make it clearer.

Changes: Lines 182,184,185

  1. “Implant lost integration after abutment fracture”: what’s the meaning behind this phrase?

Response: The authors thank the Reviewer’s query. The authors meant that the patient chief complain that brought him to the clinic was a slight mobility on the prosthesis, being diagnosed first as an abutment fracture and immediately after with implant mobility. The authors removed the comment from the table to ake it clearer for the reader.

Changes: Table 2

  1. Line 257: I would eliminate this phrase altogether an implant failure is obligatory cause of the failure of the related prostheses.

Response: The authors thank the Reviewer’s indication. The authors rephrased to “Despite the implant failures, all prostheses remained in function.” This is the meaning intended by the authors.

Changes: Line 266

  1. Line 275: Authors should specify if those complications were mucositis or peri-implantitis.

Response: The authors thank the Reviewer’s indication. The authors defined peri-implant pathology between lines 201 and 204.

Changes: None.

Reviewer 2 Report

I’ve review the manuscript titled ‘Long-term outcome of implants inserted in immediate function on autogenous grafted bone’ with interest. Authors have reported data on retrospective analysis of 10-14 years clinical and radiographic evaluation of dental implants inserted in grafted bone with immediate loading in full-arch rehabilitation of the maxillae. I found the study very interesting. Please find my comments and suggestion below.

·      Authors are encourage to compare their finding with the immediate function implants placed in nongrafted bone too, as this will also provide a useful information about the implants success on both grafted and non-grafted scenarios. 

·      Line 110-120, Please provide a table of four groups according to the patients characteristics for better understanding of the methodology of immediate prosthetic rehabilitation protocols

·      Would you please discuss further about any corelation between Figure 4 and 5 regarding implant cumulative survival rate at 14 years of follow-up and considering the patient as unit of analysis?

·      Table 1 has too many details and variable at one place, please split it for better understanding 

·      What do you think about the higher incidence of mechanical complications than biological complication?

Author Response

The authors thank the Reviewer for all the effort put into the analysis of our manuscript. Please find below the point-by-point response to the review.

I’ve review the manuscript titled ‘Long-term outcome of implants inserted in immediate function on autogenous grafted bone’ with interest. Authors have reported data on retrospective analysis of 10-14 years clinical and radiographic evaluation of dental implants inserted in grafted bone with immediate loading in full-arch rehabilitation of the maxillae. I found the study very interesting. Please find my comments and suggestion below.

1·      Authors are encourage to compare their finding with the immediate function implants placed in nongrafted bone too, as this will also provide a useful information about the implants success on both grafted and non-grafted scenarios. 

Response: The authors thank the Reviewer’s suggestion. The discussion was provided as requested.

Changes: Lines 317-326 and 337-341

References:

Del Fabbro M, Pozzi A, Romeo D, de Araújo Nobre M, Agliardi E. Outcomes of Fixed Full-Arch Rehabilitations Supported by Tilted and Axially Placed Implants: A Systematic Review and Meta-Analysis. Int J Oral Maxillofac Implants. 2022 Sep-Oct;37(5):1003-1025. Doi: 10.11607/jomi.9710.

Trbakovic A, Toljanic JA, Kumar VV, Thor A. Eight to eleven-year follow-up of immediately loaded implants placed in edentulous maxillae with compromised bone volume and poor bone quality: A prospective cohort study. Clin Implant Dent Relat Res. 2020 Feb;22(1):69-76. doi: 10.1111/cid.12874.

Cassetta M. Immediate loading of implants inserted in edentulous arches using multiple mucosa-supported stereolithographic surgical templates: a 10-year prospective cohort study. Int J Oral Maxillofac Surg. 2016 Apr;45(4):526-34. doi: 10.1016/j.ijom.2015.12.001.

Maló P, de Araújo Nobre M, Lopes A, Ferro A, Nunes M. The All-on-4 concept for full-arch rehabilitation of the edentulous maxillae: A longitudinal study with 5-13 years of follow-up. Clin Implant Dent Relat Res. 2019 Aug;21(4):538-549. doi: 10.1111/cid.12771. Epub 2019 Mar 28.

2·      Line 110-120, Please provide a table of four groups according to the patients characteristics for better understanding of the methodology of immediate prosthetic rehabilitation protocols

Response: The authors thank the Reviewer’s indication. The Table was provided.

Changes: Table 1; line 121

3·      Would you please discuss further about any corelation between Figure 4 and 5 regarding implant cumulative survival rate at 14 years of follow-up and considering the patient as unit of analysis?

Response: The authors thank the Reviewer’s query. The patient as unit of analysis is intended to provide information on how many patients were able to be free of implant failure during the follow-up of the study. In this situation, 76.8% of the patients were able to cumulatively go through the follow-up without implant failures, and therefore, 23.2% of the patients cumulatively experienced at least one implant failure. The authors included several remarks to make it clearer for the reader.

Changes: Lines 247 and 248, Table 3, Figure 4

4·      Table 1 has too many details and variable at one place, please split it for better understanding 

Response: The authors thank the Reviewer’s indication. The table was split in two: The information on Table 1- “Overall medical status distribution according to the International Classification of Disease, version 11 (ICD-11)” was inserted in a new table 1 further illustrating the sample characteristics according to the prosthetic rehabilitation group; Table 2- ” Implant distribution in the sample and Implant failure analysis “.

Changes: Table 1 (new table) in line 121

5·      What do you think about the higher incidence of mechanical complications than biological complication?

Response: The authors thank the Reviewer’s query. The higher rate of mechanical complications compared to biological complications is something that the authors are used to see on their data. The first aspect to consider is that the authors report on both provisional and definitive prostheses and not just definitive prostheses as a significant amount of literature. This means that almost half of the reported mechanical complications (45%) are reported in the provisional prosthesis. A second aspect, biological complications can be prevented in a significant amount by a good maintenance recall program (despite most times the patients don’t feel anything until the pathology is in an advanced state). However, Mechanical complications are more difficult to prevent as it is more sensitive and influenced by several factors: The patient’s occlusion that might change slightly due to trauma or wearing of the crowns (that they don’t feel) that might mean a dramatic change in load distribution, if the patient uses the night guard daily as prescribed or only sometimes, if the patient has an implant-supported fixed prosthesis as opposing dentition that would mean no shock absorbing characteristics in occlusion when compare for example to natural teeth as opposing dentition (reverting to the previous statement about the night guard); and the materials that also have their limits on the long term. These are probably the most common reasons for the high incidence of mechanical complications. However, given that it was not possible to collect information on some factors, the authors could not include these explanations but rather recognize the study limitations. We do appreciate the Reviewer allowing us to further elaborate on our opinion.

Changes: None.

Round 2

Reviewer 1 Report

All my previous comments were properly replied to.